# A Preliminary Comparison of Plasma Tryptophan Metabolites and Medium- and Long-Chain Fatty Acids in Adult Patients with Major Depressive Disorder and Schizophrenia

**DOI:** 10.3390/medicina59020413

**Published:** 2023-02-20

**Authors:** Jun-Chang Liu, Huan Yu, Rui Li, Cui-Hong Zhou, Qing-Qing Shi, Li Guo, Hong He

**Affiliations:** 1Department of Psychiatry, Gaoxin Hospital, Xi’an 710077, China; 2Department of Psychiatry, Xijing Hospital, Air Force Medical University, Xi’an 710032, China; 3Department of Psychiatry, Chang’an Hospital, Xi’an 710100, China

**Keywords:** tryptophan metabolites, medium- and long-chain fatty acids, plasma, major depressive disorder, schizophrenia

## Abstract

*Background and Objectives*: Disturbance of tryptophan (Trp) and fatty acid (FA) metabolism plays a role in the pathogenesis of psychiatric disorders. However, quantitative analysis and comparison of plasma Trp metabolites and medium- and long-chain fatty acids (MCFAs and LCFAs) in adult patients with major depressive disorder (MDD) and schizophrenia (SCH) are limited. *Materials and Methods*: Clinical symptoms were assessed and the level of Trp metabolites and MCFAs and LCFAs for plasma samples from patients with MDD (*n* = 24) or SCH (*n* = 22) and healthy controls (HC, *n* = 23) were obtained and analyzed. *Results*: We observed changes in Trp metabolites and MCFAs and LCFAs with MDD and SCH and found that Trp and its metabolites, such as N-formyl-kynurenine (NKY), 5-hydroxyindole-3-acetic acid (5-HIAA), and indole, as well as omega-3 polyunsaturated fatty acids (N3) and the ratio of N3 to omega-6 polyunsaturated fatty acids (N3: N6), decreased in both MDD and SCH patients. Meanwhile, levels of saturated fatty acids (SFA) and monounsaturated fatty acids (MUFA) decreased in SCH patients, and there was a significant difference in the composition of MCFAs and LCFAs between MDD and SCH patients. Moreover, the top 10 differential molecules could distinguish the two groups of diseases from HC and each other with high reliability. *Conclusions*: This study provides a further understanding of dysfunctional Trp and FA metabolism in adult patients with SCH or MDD and might develop combinatorial classifiers to distinguish between these disorders.

## 1. Introduction

Major depressive disorder (MDD) and schizophrenia (SCH) are severe psychiatric diseases that affect a large number of people worldwide [1,2]. These two diseases’ diagnoses are still mainly defined by their clinical features. However, converging evidence suggests that these disorders have considerable overlap in symptoms and heritage patterns [3,4,5], making it sometimes difficult to distinguish between SCH and MDD. Therefore, clinical characteristics alone are of limited predictive value and biological predictors will be of enormous value [6].

Tryptophan (Trp) and its metabolites are involved in the regulation of neuronal function and immunity, and the imbalances in Trp metabolism have resulted in neurodegenerative disease [7]. A significant amount of evidence suggests that plasma Trp and its catabolites have been considered contributing factors in the etiology of MDD and SCH [8,9]. On the other hand, fatty acids (FAs) also play a role in neural membrane fluidity and receptor binding, affecting neurological functions such as synthesizing and releasing neurotransmitters, neurogenesis, and myelination [10,11,12]. The abnormal composition of FAs in the plasma, especially the changes of omega-3 (N3) polyunsaturated fatty acids (PUFAs) and omega-6 (N6) PUFAs in patients with MDD and SCH have been widely reported [13,14]. Although several Trp metabolites and FAs might be potential peripheral biomarkers for MDD and SCH [15,16,17], thus far, no studies have directly compared the plasma Trp metabolites and FA compositions between adult patients with SCH and MDD, so the development of combinational molecular biomarkers might be useful in distinguishing these two diseases.

In the present study, we performed a case-control study using liquid chromatography- or gas chromatography-mass spectrometry (LC-MS or GC-MS)-based analysis of plasma samples (*n* = 69) from sex- and age-matched adult individuals with MDD (*n* = 24), SCH (*n* = 22) and healthy controls (HC, *n* = 23). We sought to determine the differences in Trp metabolites and the composition of medium- and long-chain fatty acids (MCFAs and LCFAs) and analyze the correlation between differential profiles and clinical symptoms. Moreover, we also sought to identify discriminative combined panels that can distinguish individuals with MDD, individuals with SCH, and HCs using random forest and receiver operating characteristic (ROC) analysis.

## 2. Materials and Methods

### 2.1. Subjects and Plasma Sampling

This study was performed in accordance with the tenets of the Declaration of Helsinki. All subjects volunteered to participate in this study and provided written informed consent. MDD and SCH were diagnosed based on a Structured Clinical Interview for Diagnostic and Statistical Manual (DSM-5) of Mental Disorders criteria by two senior psychiatrists. The Mini-International Neuropsychiatric Interview was used to screen for preexisting psychiatric disorders. The Hamilton Anxiety Scale (HAMA), Hamilton Depression Rating Scale (HAMD), and Positive and Negative Syndrome Scale (PANSS) were independently administered by two psychiatrists who were blinded to the clinical status of the participants. The scores were positively correlated with anxiety, depression, and psychosis symptoms, respectively. The exclusion criteria were obesity, defined as a body mass index (BMI) ≥ 28.0; hypertension; a high-fat diet or vegetarian; lactation, or menstruation; alcohol abuse or dependence; illicit drug use; and presence of other mental disorders according to DSM-5 criteria. Finally, 24 patients with MDD (average age 33, 8 male and 16 female) and 22 patients with SCH (average age 31, 7 male and 15 female) were recruited from the Department of Psychiatry at Xijing, Gaoxin and Chang’an Hospital, along with 23 HCs (average age 29, 7 male and 16 female), all of whom underwent a physical examination. Blood samples were collected in anticoagulant tubes and centrifuged at 1600 rpm for 15 min between 8 AM and 10 AM from all participants under fasting conditions. The obtained plasma was stored in sterile cryopreservation tubes and stored in liquid nitrogen until further analysis.

### 2.2. Detection of Tryptophan and Its Catabolites

Take out the samples and slowly dissolve at 4 °C, then take 200 μL of each sample and add 800 μL of precooled methanol acetonitrile solution (1:1, *v*/*v*), vortex for 60 s, and place at −20 °C for 1 h to precipitate the protein. Then, centrifuge at 4 °C for 20 min (14,000 rcf), take the supernatant, and freeze-dry. Separate the samples using the Agilent 1290 Infinity UHPLC system. Place the standard in the 4 °C automatic samplers. The column temperature is 50 °C, the flow rate is 400 μL/min, and the injection volume is 5 μL. The relevant liquid phase gradient is as follows: 0–2 min, 15% for solution B; linear change from 15% to 98% in 2–9 min; 9–11 min, 98% for solution B; 11–11.5 min, solution B changes linearly from 98% to 15%; 11.5–14 min, 15% for liquid B. The 5500 QTRAP mass spectrometer (AB SCIEX) is used for mass spectrometry analysis in positive ion mode. 5500 QTRAP ESI source conditions are as follows: source temperature: 550 °C; Ion Source Gas1 (Gas1): 55; Ion Source Gas2 (Gas2): 55; Curtain gas (CUR): 40; ionSapary Voltage Floating (ISVF): +4500 V. Use MRM mode to detect ion pairs to be measured and MultQuant or Analyst software for the quantitative analysis.

### 2.3. Detection of Medium and Long-Chain Fatty Acids

Thaw the sample on ice and take 150 μL of the sample in a 2 mL glass centrifuge tube. Add 1 mL of chloroform-methanol solution, ultrasound for 30 min, take the supernatant, add 2 mL of 1% sulfuric acid methanol solution and put it on a water bath at 80 °C, methylate it for half an hour, add 1 mL of N-hexane for extraction, add 5 mL of pure water for washing, and suck 500 μL of the supernatant. Then, add 25 μL of methyl salicylate as the internal standard, mix and add it into the injection bottle, and then test using GC-MS. The injection volume is 1 μL and the split ratio is 10:1, split injection. The samples are collected on Agilent DB-WAX capillary column (30 m × 0.25 mm ID × 0.25 μm) gas chromatography system. Programmed temperature rise: initial temperature 50 °C; keep for 3 min, and raise the temperature to 220 °C at 10 °C/min and maintain for 5 min. The carrier gas is helium, and the carrier gas flow rate is 1.0 mL/min. One QC sample shall be set for a certain number of experimental samples at every interval in the sample queue to detect and evaluate the stability and repeatability of the system. The Agilent 7890/5975C GC-MS is used for mass spectrometry analysis. The injection port temperature is 280 °C; the ion source temperature is 230 °C; the transmission line temperature is 250 °C and uses an electron impact ionization (EI) source, SIM scanning mode, and electron energy of 70 eV. Use MSD ChemStation software (MSD ChemStation, Agilent Technologies, Santa Clara, CA, USA) to extract the chromatographic peak area and retention time. Draw the calibration curve and calculate the content of MCFAs and LCFAs in the sample.

### 2.4. Statistical Analyses

Statistical analyses were performed using SPSS 19.0 software (IBM-SPSS Inc., Chicago, IL, USA) and R-4.0.5 (R Core Team, Vienna, Austria). Differences in continuous variables were assessed using the Kruskal–Wallis test (abnormal distribution) or one-way analysis of variance combined with Bonferroni correction (normal distribution). The measurement data conforming to the normal distribution is represented by mean ± SD and the measurement data nonconforming to the normal distribution is represented by M (P25, P75). Comparison of counting data was expressed in number and percentage and assessed using χ2 test. *p* < 0.05 indicates that the difference is statistically significant. The Spearman correlation analysis was used to assess the correlations between the clinical parameters and differential metabolites. To obtain simplified potential biomarker panels, the online software MetaboAnalyst 5.0 (https://www.metaboanalyst.ca/, accessed on 12 October 2022) was used to conduct random forest analysis, screen biomarkers, and draw ROC curves, and the metabolite of Top10 was selected as the candidate biomarker.

## 3. Results

### 3.1. Clinical Characteristics of the Recruited Participants

A total of 69 individuals were included in the study. No significant differences were found among the three groups in terms of age (*p* = 0.288), gender (*p* = 0.978), and BMI (*p* = 0.058). Scores of HAM-D and HAM-A in the MDD group were higher than those in the SCH and HC group (Table 1). PANSS total score (T), positive symptom score (P), negative symptom score (N), and general psychopathological symptom score (G) in the SCH group and PANSS (T), PANSS (N) and PANSS (G) in the MDD group were higher than those in the HC group. Moreover, PANSS (P) and PANSS (N) in the SCH group were also higher than those in the MDD group. There was a significant difference among the three groups in terms of marital status, and there was no significant difference in terms of smoking (*p* = 0.546).

### 3.2. Alternation of Trp and Its Catabolites in SCH and MDD

A total of 11 Trp metabolites were identified. There were significant differences in the levels of tryptophan (Try, *F* = 17.10, *p* < 0.001) (Figure 1A), 5-hydroxyindole-3-acetic acid (5-HIAA, *F* = 5.551, *p* < 0.001) (Figure 1B), N-formyl-kynurenine (NKY, *F* = 3.551, *p* = 0.034) (Figure 1C), Indole (*F* = 38.36, *p* < 0.001) (Figure 1D), Indole-3-carboxaldehyde (IAId, *F* = 18.80, *p* < 0.001) (Figure 1E), Indoleacetate (IAA, *F* = 5.422, *p* < 0.01) (Figure 1F), and indole-3-lactic acid (ILA, *F* = 28.12, *p* < 0.001) (Figure 1G) among the three groups. Intercomparison further showed that the levels of Try, 5-HIAA, NKY, Indole, IAId, and ILA decreased in both MDD and SCH groups compared with the HC group. Meanwhile, levels of IAA decreased in the MDD group compared with the HC group (Figure 1F), and levels of ILA also decreased in the MDD group compared with the SCH group (Figure 1G). Furthermore, levels of IAId, Indole, ILA, Try, 5-HIAA, NKY, and IAA were negatively correlated with the scores of HAMA, HAMD, and PANSS (T) (Figure 1H). However, there were no significant differences in the levels of L-kynurenine, picolinic acid, quinolinic acid, and 3-indoxyl sulfate among the three groups (Table 2). These results suggested that levels of plasma Trp and its catabolites in the SCH and MDD groups were largely changed, but only ILA is the differential metabolite between these two groups.

### 3.3. Alternation of MCFAs and LCFAs in SCH and MDD

A total of 35 medium- and long-chain fatty acids were identified in samples from each group. GC-MS analysis revealed that there were significant differences in the levels of saturated fatty acids (SFA, *F* = 4.773, *p* = 0.012), monounsaturated fatty acids (MUFA, *F* = 3.736, *p* = 0.029), omega-3 polyunsaturated fatty acids (N3, *F* = 18.028, *p* < 0.001), and the N3:N6 ratio (*F* = 7.748, *p* = 0.001), whereas there were no significant differences in the levels of total FAs (*F* = 0.991, *p* = 0.377), polyunsaturated fatty acids (PUFA, *F* = 2.767, *p* = 0.070), and omega-6 polyunsaturated fatty acids (N6, *F* = 3.041, *p* = 0.055) (Figure 2A–G). Meanwhile, levels of N3 and the N3/N6 ratio in both MDD and SCH groups were less than those in HC group. The SCH group also showed decreased SFA and MUFA compared with the HC group. Otherwise, levels of N3 were negatively correlated with the scores of HAMA, HAMD, PANSS (T), and PANSS (N), whereas the N3/N6 ratio was negatively correlated with the scores of PANSS (T) and PANSS (P) (Figure 2H). There were also significant differences in the levels of multiple fatty acids, including C8:0 (*F* = 33.082, *p* < 0.001), C10:0 (*F* = 6.664, *p* = 0.002), C12:0 (*F* = 4.875, *p* = 0.011), C13:0 (*F* = 16.166, *p* < 0.001), C18:0 (*F* = 6.172, *p* =0.004) and C20:0 (*F* = 5.872, *p* = 0.004) in SFA; C14:1N5 (*F* = 4.551, *p* = 0.014), C22:1N9 (*F* = 18.179, *p* < 0.001) and C24:1N9 (*F* = 15.481, *p* < 0.001) in MUFA; C22:6N3 (*F* = 11.865, *p* < 0.001), C22:5N3 (*F* = 4.083, *p* = 0.021), C18:2N6 (*F* = 3.354, *p* = 0.041), C18:3N6 (*F* = 14.976, *p* < 0.001), C20:4N6 (*F* = 3.307, *p* = 0.043), C22:2N6 (*F* = 9.249, *p* < 0.001), and C20:4N6 (*F* = 3.651, *p* = 0.031) in PUFA. Intercomparison further showed that the levels of C8:0, C10:0, C12:0, C13:0, C22:1N9, C22:6N3, and C22:5N3 decreased, whereas those of C14:1N5, C24:1N9, C18:3N6, C22:4N6, and C22:2N6 increased in the MDD group compared with the HC group. Meanwhile, levels of C8:0, C10:0, C13:0, C18:0, C24:1N9, C20:5N3, C22:6N3, and C22:5N3 decreased, whereas those of C20:0, C14:1N5, C18:2N6, C18:3N6, and C20:4N6 increased in the SCH group compared with the HC group. Moreover, levels of C18:0, C24:1N9, and C22:2N6 increased, whereas C20:0 and C22:1N9 decreased in the MDD group compared with the SCH group (Table 3).

### 3.4. Characteristic Fatty Acids and Trp Catabolites in SCH and MDD

Based on random forest analysis, the top 10 differential molecules, including C22:1N9, indole, C8:0, C10:0, ILA, C18:3N6, IAId, Trp, C13:0, and NKY, were selected as potential biomarker panel, which could effectively distinguish between MDD and HC (AUC = 0.99, Figure 3A,B). Moreover, a panel containing C8:0, C24:1N9, indole, C18:3N6, NKY, C10:0, C13:0, IAId, QUIN, and C24:0 could effectively distinguish between SCH and HC (AUC = 0.996, Figure 3C,D). Notably, a panel consisting of C24:1N9, C22:1N9, C22:2N6, ILA, C18:1N9, C22:0, C16:1N7, C18:0, IAId, and C20:2N6 could effectively distinguish between MDD and SCH (AUC = 0.981, Figure 3E,F).

## 4. Discussion

In this study, we investigated the compositions of plasma Trp metabolites and MCFAs and LCFAs in adult patients with MDD and SCH. We found that Trp and its metabolites such as 5-HIAA, NKY, and indole, and total N3 and the N3:N6 ratio decreased in both the MDD and SCH groups. Intriguingly, the composition of MCFA and LCFA was different between MDD and SCH, while only ILA in tryptophan metabolites was different between these two groups. Moreover, the top 10 differential molecules, which could distinguish the two groups of diseases from HC and each other with high reliability, were screened. These results may shed light on the investigation of diagnostic molecular targets for SCH and MDD and are worth further exploration due to the limited sample size.

Trp is an essential amino acid and a biosynthetic precursor of a large number of metabolites [18]. In the gastrointestinal tract, Trp derivatives are formed in three main pathways: the kynurenine pathway (KP), the major metabolic pathway for free Trp, which leads to the formation of several metabolites with distinct biological activities in the neurotransmission and immune response such as NKY, L-kynurenine, kynurenic acid (KYNA), picolinic acid, and quinolinic acid [19]; the serotonin production pathway via Trp hydroxylase 1, which produces serotonin and could further metabolize into 5-HIAA [20]; and the microbial metabolism pathway, which directly transforms Trp into several molecules such as indole, IAA, indoxyl sulfate (IS), and IAId [21]. The rate-limiting step is the conversion of Trp to N-formyl-kynurenine (NKY) by indoleamine-2,3-dioxygenase (IDO) and tryptophan-2,3-dioxygenase (TDO). The abnormalities of the Trp metabolites, especially in KP, have been observed in MDD. For instance, plasma Trp and KYNA decreased in patients with MDD [22,23]. Correspondingly, IDO activity was elevated in MDD and positively correlated with depressive symptoms [24]. Consistent with these results, the present study found that Trp and NKY decreased in patients with MDD, indicating that Trp metabolism increased in MDD. However, we did not observe changes in QUIN and PIC in patients with MDD compared with HCs. QUIN is considered a neurotoxic metabolite that participates in the generation of ROS and stimulates synaptosomal glutamate release [25], whereas PIC is considered a neuroprotective metabolite that exhibits immunomodulatory properties [26]. Whereas several works found that peripheral QUIN increased [27] while PIC decreased in patients with MDD [28,29], a recent study comparing MDD cases versus controls found that there were no significant differences in tryptophan catabolites [9]. On the other hand, decreased Trp and elevated kynurenine/Trp ratios were also reported in patients with SCH [16]. Similarly, the present study also found that Trp and NKY decreased in patients with SCH, indicating that Trp metabolism increased in both MDD and SCH patients. Of note, the present study did not observe changes in QUIN and PIC in patients with SCH compared with HCs. Although a previous study found that peripheral QUIN was in patients with SCH [30], a systematic review inferred that there was no significant increase in QUIN and PIC, and peripheral blood levels of tryptophan catabolites were dissociated from central nervous system findings except for a modest increase in the serum IDO activity of patients with SCH [31]. Due to effect sizes varying greatly between studies assessing kynurenine pathway metabolites in MDD and SCH groups [23,32], a longitudinal trial with a large sample size might further clarify these differences.

Alternation of 5-HIAA in patients with MDD and SCH has also been reported. A previous study found that plasma 5-HIAA increased in depressed patients [33]. However, a recent systematic umbrella review indicated the hypothesis that depression is caused by lowered serotonin activity or concentrations is not convincing enough, and 5-HIAA concentrations in cerebrospinal fluid are not associated with depression [34]. Intriguingly, another work found that plasma 5-HIAA levels are negatively correlated with the depression/anxiety component in patients with SCH [35]. The present found that 5-HIAA decreased in patients with MDD and SCH and was negatively correlated with the scores of HAMA, HAMD, PANSS (N), and PANSS (T), indicating that plasma 5-HIAA might be related to emotional symptoms rather than disease types. Furthermore, indole metabolites, such as IAId, ILA, and IAA, are considered beneficial to inhibit neuronal damage and exert anti-inflammatory effects [36,37]. Importantly, lower concentrations of Trp and indoles, particularly IAld in serum, are correlated with more severe depressive symptoms [38]. In line with this study, we found that indole, IAId, and IAA decreased in both MDD and SCH patients and were negatively correlated with the scores of HAMA, HAMD, and PANSS (T), suggesting that the unbalance of indole metabolism might be a common characteristic of these two diseases.

Medium- and long-chain fatty acids (MCFAs and LCFAs) are natural compounds that mainly participate in cell metabolism. MCFAs are important food constituents that contain total carbon atom numbers from 6 to 12 [39], whereas those greater than 12 are considered long-chain fatty acids. The roles of MCFAs within gluconeogenesis and lipogenesis as well as mitochondrial function and metabolism have been uncovered currently [40,41]. Although animal studies have revealed that exogenous supplementation of MCFAs or their esters (medium-chain triglycerides, MCT) can improve depressive behavior and cognitive function [42], so far, changes in peripheral MCFAs in patients with depression and schizophrenia have been poorly investigated. In the present study, levels of caprylic acid (C8:0) decreased in both MDD and SCH patients and were negatively correlated with the scores of HAMA, HAMD, PANSS (N), and PANSS (T). Meanwhile, capric acid (C10:0) and lauric acid (C12:0) decreased in patients with MDD and were negatively correlated with the scores of HAMA or scores of HAMA and HAMD, respectively. Given the neuroprotective effects and neuroregulation of caprylic acid, capric acid, and lauric acid [43,44], their potential role in the pathogenesis of neurological disorders still needs to be further determined.

In the present study, we further compare the concentrations of SFAs, MUFAs, and PUFAs in HC, MDD, and SCH. SFAs have been shown to influence several brain circuits and thus regulate mood [45]. SFAs could stimulate the release of pro-inflammatory cytokines and induce the apoptosis of astrocytes [46]. Whereas intake of SFAs causes impairments in the activity of the brain dopamine system and induces depressive-like behavior in rodents [47,48], the accumulation of SFAs may play a role in the pathogenesis of MDD and SCH. In contrast with the effects of SFAs, pre-clinical findings reveal that the intake of MUFAs has been suggested to improve brain function, such as the protection of the integrity of the dopamine system and facilitation of neurotransmitter signal transduction [49,50]. Importantly, a clinical study also found that a MUFA-enriched diet in humans reduces the risk of depression [51]. In the present study, although there is no significant difference between MDD and HC in the plasma total SFA and MUFA, C22:1N9 decreased and C14:1N5 and C24:1N9 increased in patients with MDD compared with HCs or patients with SCH. Meanwhile, C8:0, C10:0, C12:0, and C13:0 decreased in patients with MDD compared with HCs. On the contrary, plasma total SFA and MUFA decreased in patients with SCH compared with HCs. Notably, we found that C18:0, C16:1N7, and C20:1N9 decreased in patients with SCH compared with HCs or patients with MDD. Moreover, we also found that C8:0, C10:0, C13:0, C18:0, and C24:1N9 decreased whereas those of C20:0 and C14:1N5 increased in the SCH group compared with the HC group. Intriguingly, levels of C18:0, C24:1N9, and C22:2N6 were increased whereas C20:0 and C22:1N9 decreased in the MDD group compared with the SCH group. Therefore, the changes in plasma SFA and MUFA in patients with SCH are more than those in patients with MDD, and the abnormal composition of SFA and MUFA may be one of the peripheral mechanisms leading to depressive and psychiatric symptoms.

PUFAs play several important roles in brain function and neural diseases, such as regulation of neurotransmission, neuroinflammation, mood, and cognition [52]. Previous studies mainly focused on N3 and N6 PUFAs in samples of patients with MDD. Although previous studies provide evidence of decreased N3 PUFA and changed N6 PUFA levels in patients with MDD [53], the link between N-3/N-6 PUFA and MDD is inconsistent [54]. In confirmation with these results, we found that levels of N3 as well as DHA and DPA, and the N3:N6 ratio decreased in patients with MDD. However, there is no significant difference between HC and MDD on the levels of N6 and total PUFA, indicating that decreased plasma N3 is a feature of MDD. Altered N3 and N6 PUFAs in samples of patients with SCH have also been reported previously [55]. Recent work has indicated that long-chain N3 and N6 concentrations are associated with a lower risk of schizophrenia. By contrast, there is weak evidence that short-chain N3 and N6 are associated with an increased risk of schizophrenia [56]. The present study also found that N3, as well as EPA, DHA, and DPA and the N3:N6 ratio, decreased in patients with SCH, indicating that decreased plasma N3 is a common feature of the two diseases. Of note, a number of studies have found that N3 but not N6 decrease in patients with MDD. For example, levels of N3 were significantly lower in depressive patients and there was no significant change in N6 between patients with MDD and control subjects [53], and lower levels of total N3 and increased N6/N3 ratios were also reported in perinatal depression patients [57]. However, a recent study found that N3 is lower in depression, but it is not consistently associated with subsequent change in depressive symptoms [58]. Meanwhile, circulating PUFAs are unlikely to reflect a vulnerability marker for the recurrence of depression [59]. Another study further indicated that the N6:N3 ratio is positively associated with MDD at age 24, but there was little evidence of cross-sectional associations between PUFA measures and mental disorders at age 17 [60]. On the other hand, dietary supplementation with N3 during pregnancy or postpartum reduces depressive symptoms and N3 adjuvant treatment is a potential option for depression and anxiety symptoms of people with recent onset psychosis [61,62]. However, recent studies found that N3 have an overall significant beneficial effect on perinatal depression [63], but the evidence of N3 supplementation on sertraline continuous therapy to reduce depression or anxiety symptoms is not enough to make recommendations [64]. Moreover, other studies have also found that N3 supplementation probably has little or no effect in preventing depression or anxiety symptoms [65] and a non-clinical beneficial effect on depressive symptomology compared to placebo in adult patients with depression [66]. Furthermore, basic study has already shown that complementation of diet with arachidonic acid (AA, C20:4N6) is sufficient to alleviate both the microbiota-induced depressive-like behaviors [67]. Together, the changes of PUFA in patients with depression in different studies are not completely consistent, and the results of this study need to be further verified by a large sample size cohort study.

Additionally, we screened the combinatorial markers of Trp metabolites and FAs through random forest analysis. Among them, C8:0, C10:0, C13:0, C18:3N6, indole, NKY, and IAId were the common characteristic molecules in both MDD and SCH, indicating that these disorders have overlapping pathogenesis, which may also be one of the potential mechanisms behind the existence of similar symptoms in both MDD and SCH groups [3,4]. Moreover, the combinatorial markers which can distinguish between MDD and SCH mostly come from fatty acids, suggesting that fatty acid metabolism may play a role in distinguishing between the two diseases.

Finally, several potential limitations should be mentioned. Because Trp and fatty acids metabolism is greatly affected by individuals, such as gender, age, eating habits, and smoking [7,68], the recruited cases are relatively small and the results still need a large sample to explore further. Meanwhile, considering that antidepressant and antipsychotic drugs could affect Trp and fatty acids metabolism [69,70,71], the influence of drugs on plasma Trp metabolites and the composition of fatty acids cannot be excluded. In addition, the detection methods based on mass spectrometry require expensive equipment and may lead to discrepancies in results. Several important molecules related to psychiatric disorders in the plasma, such as 5-HT and melatonin, were not detected in the present study. Furthermore, several fatty acid contents detected in the present study were not consistent with previous work. For example, the concentrations of C14:0, C18:0, and C22:0 was inconsistent with those in other studies [72,73]. This inconsistency may be related to the participants’ eating habits, living standards, gender, and age. Of note, the influence of plasma collection methods, storage conditions, and fatty acid detection and analysis methods on the results cannot be ignored [74]. Therefore, the detection accuracy needs to be improved, and its clinical application is also limited at this stage. Nevertheless, both fatty acid and Trp metabolism are regulated by gut microbiota, and the metabolic pathway of fatty acids in central nervous system diseases and the composition characteristics of gut microbiota in depression and schizophrenia have been clarified [50,75,76]. The potential metabolic pathway of fatty acids in patients with MDD and SCH and the relationship between specific gut microbiota, fatty acid metabolism, and tryptophan metabolism also need to be further explored.

## 5. Conclusions

In summary, the present study characterized plasma Trp metabolites and the composition of MCFAs and LCFAs in adult patients with MDD and SCH. Trp and NKY as well as N3 and the N3:N6 ratio decreased in both patients with MDD and those with SCH. Furthermore, combinatorial classifiers that could discriminate MDD from HC or SCH from HC, as well as MDD from SCH, were developed. Intriguingly, UFA may play a role in distinguishing between MDD and SCH. Our findings might provide a potential strategy to combine Trp with FA metabolism for the diagnosis of MDD and SCH.

## Figures and Tables

**Figure 1 medicina-59-00413-f001:**
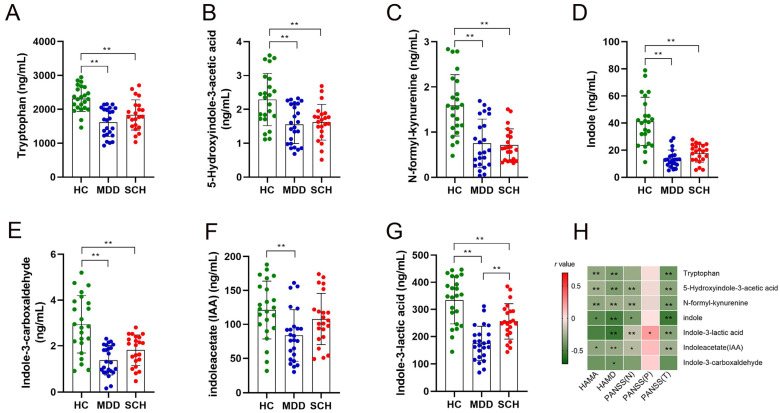
Comparison of concentrations of differential Trp metabolites among HC, MDD, and SCH participants. (**A**) Tryptophan, (**B**) 5-hydroxyindole-3-acetic acid, (**C**) *N*-formyl-kynurenine, (**D**) Indole, (**E**) Indole-3-carboxaldehyde, (**F**) Indoleacetate, (**G**) Indole-3-lactic acid, and (**H**) correlation between clinical parameters and changed metabolites. Red squares show a positive correlation and green squares show a negative correlation, * *p* < 0.05; ** *p* < 0.01.

**Figure 2 medicina-59-00413-f002:**
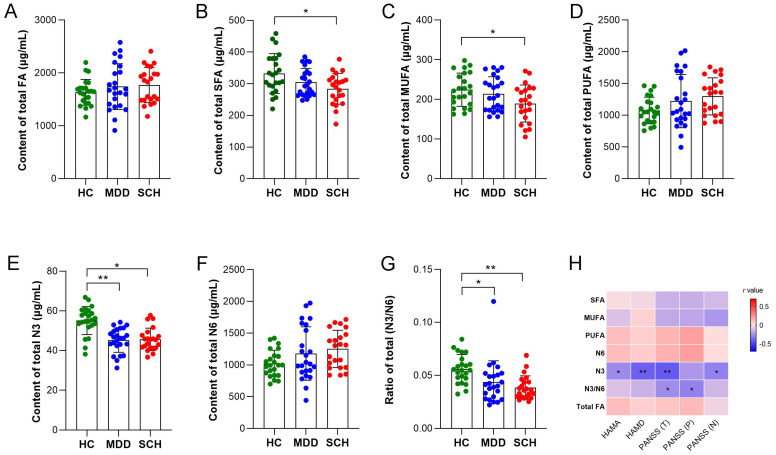
Comparison of concentrations of medium- and long-chain fatty acids among HC, MDD, and SCH participants. Comparison of the concentration of total FA (**A**), total SFA (**B**), total MUFA (**C**), total PUFA (**D**), total N3 (**E**), total N6 (**F**) and the ratio of N6/N3 (**G**). (**H**) Correlation between clinical parameters and changed fatty acids. Red squares show a positive correlation and blue squares show a negative correlation, * *p* < 0.05; ** *p* < 0.01. Abbreviations: FA, fatty acids; SFA, saturated fatty acids; MUFA, monounsaturated fatty acids; PUFA, polyunsaturated fatty acids; N3, omega-3 polyunsaturated fatty acids; N6, omega-6 polyunsaturated fatty acids.

**Figure 3 medicina-59-00413-f003:**
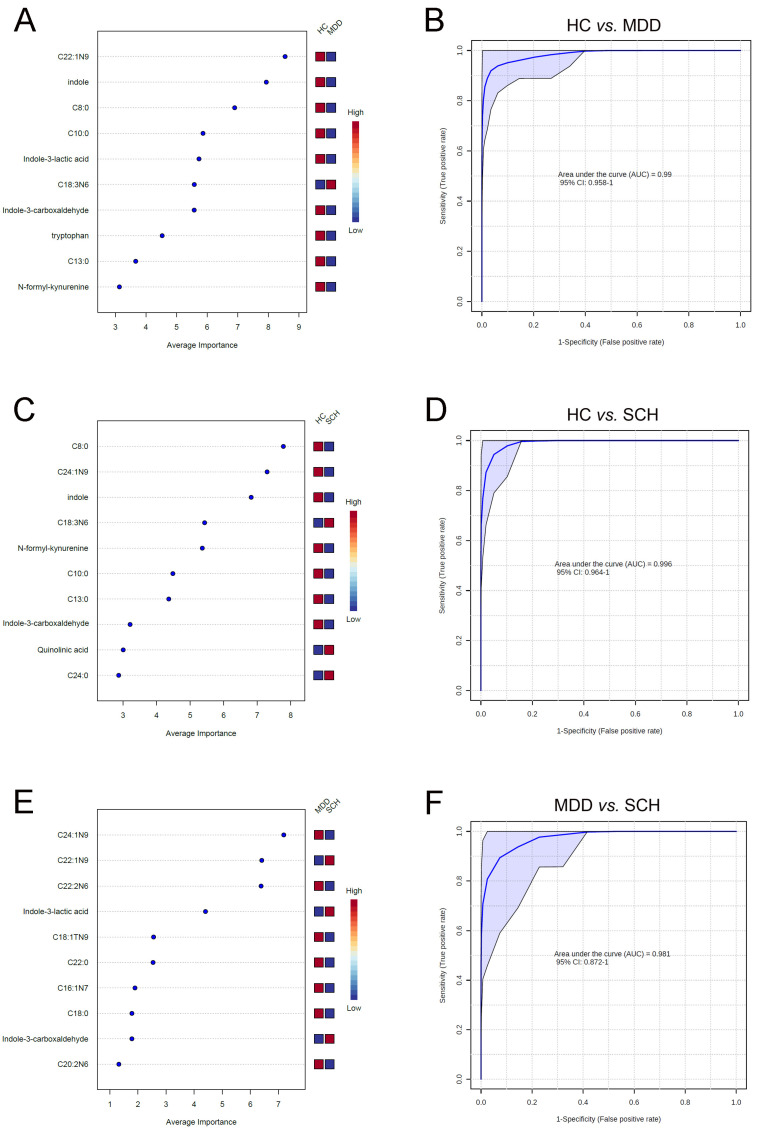
Metabolomic markers and AUC value of ROC analysis for discriminating between MDD, SCH, and HC. Top 10 differential markers identified from random forest classifiers based on the combination of Trp and fatty acids and AUC value of ROC analysis to detect MDD from HC (**A**,**B**), SCH from HC (**C**,**D**), and MDD from SCH (**E**,**F**). Abbreviations: C8:0, caprylic acid; C10:0, capric acid; C13:0, tritridecanoin; C18:0, stearic acid; C22:0, behenic acid; C24:0, lignoceric acid; C16:1N7, palmitoleic acid; C22:1N9, erucic acid; C24:1N9, nervonic acid; C18:1TN9, elaidic acid; C18:3N6, gamma-linolenic acid; C22:2N6, docosadienoic acid; C20:2N6, eicosatrienoic acid.

**Table 1 medicina-59-00413-t001:** The comparison of clinical characteristics data and symptom scale assessment among the three groups.

Parameter	HC (*n* = 23)	MDD (*n* = 24)	SCH (*n* = 22)	*F/H/ χ^2^* Value	*p*-Value
Sociodemographic					
Age [years, M (*P*_25_, *P*_75_)] ^a^	29 (26, 32)	33 (27.25, 36)	31 (27, 34.75)	*H* = 2.488	0.288
Gender (male/female) ^c^	7/16	8/16	7/15	χ^2^ = 0.045	0.978
BMI [kg/m^2^, mean ± SD] ^b^	20.91 ± 2.78	21.01 ± 2.51	22.85 ± 3.65	*F* = 2.967	0.058
Marital status (single/married) ^c^	9/14	1/23	15/7	χ^2^ = 20.484	<0.001
Smoking (almost none/intermittent) ^c^	16/7	13/11	14/8	χ^2^ = 1.210	0.546
Scale evaluation					
HAMD (mean ± SD) ^a^	3.61 ± 1.75	21.92 ± 7.62 *	7.00 ± 2.94 *^#^	*H* = 50.719	<0.001
HAMA [M (*P*_25_, *P*_75_)] ^a^	5 (2, 6)	24 (17, 29) *	6 (4, 8) ^#^	*H* = 44.128	<0.001
PANSS (T) [ M (*P*_25_, *P*_75_)] ^a^	34 (32, 39)	58.50 (50.25, 65) *	72.5 (59.25, 83.75) *	*H* = 44.576	<0.001
PANSS (P) [M (*P*_25_, *P*_75_)] ^a^	7 (7, 8)	7 (7, 7)	18 (15.5, 25) *^#^	*H* = 47.114	<0.001
PANSS (N) [M (*P*_25_, *P*_75_)] ^a^	8 (7, 9)	11.00 (7.25, 13) *	14.00 (12, 19.25) *^#^	*H* = 29.097	<0.001
PANSS (G) (mean ± SD) ^a^	19.39 ± 2.41	40.38 ± 8.35 *	38.73 ± 10.57 *	*H* = 43.176	<0.001

Abbreviations: HC, healthy controls; MDD, Major depressive disorder; SCH, schizophrenia; ^a^ Kruskal–Wallis; ^b^ One-way analysis of variance (ANOVA); ^c^ Chi-square tests. * *p* < 0.05 vs. HC group; ^#^ < 0.05 vs. MDD group; BMI: body mass index; values are shown as mean ± SD or M (*P*_25_, *P*_75_); SD, standard deviation; PANSS (T), PANSS total score; PANSS (P), PANSS positive symptom score; PANSS (N), PANSS negative symptom score; PANSS (G), PANSS general psychopathological symptom score.

**Table 2 medicina-59-00413-t002:** Trp metabolites with no significant difference among the three groups.

Parameter (ng/mL)	HC	MDD	SCH	*F* Value	*p*-Value
l-kynurenine	108.35 ± 14.12	117.55 ± 24.81	117.23 ± 19.26	1.570	0.216
Picolinic acid	6.62 ± 1.67	7.68 ± 2.98	7.39 ± 2.43	1.197	0.309
Quinolinic acid	41.19 ± 14.63	36.36 ± 3.81	38.44 ± 4.27	1.679	0.194
3-Indoxyl sulfate	356.22 ± 185.53	317.24 ± 185.12	420.62 ± 200.26	1.856	0.164

**Table 3 medicina-59-00413-t003:** Comparison of FA among the three groups.

Parameter (μg/mL)	HC	MDD	SCH	*F* Value	*p*-Value
C8:0	0.183 ± 0.05	0.010 ± 0.003 **	0.010 ± 0.002 **	33.082	<0.001
C10:0	0.128 ± 0.001	0.009 ± 0.004 **	0.009 ± 0.004 **	6.664	0.002
C12:0	0.166 ± 0.092	0.105 ± 0.052 **	0.119 ± 0.058	4.875	0.011
C13:0	0.078 ± 0.071	0.011 ± 0.008 **	0.011 ± 0.008 **	16.166	<0.001
C14:0	3.859 ± 1.532	3.201 ± 1.300	3.310 ± 1.263	1.551	0.22
C15:0	0.943 ± 0.279	0.941 ± 0.308	0.914 ± 0.316	0.071	0.931
C16:0	223.736 ± 47.742	203.393 ± 35.148	197.029 ± 36.323	2.788	0.069
C17:0	2.307 ± 0.548	2.020 ± 0.509	2.221 ± 0.552	1.71	0.189
C18:0	96.214 ± 16.662	91.945 ± 11.846 ^#^	80.646 ± 17.381 **	6.127	0.004
C20:0	0.755 ± 0.172	0.790 ± 0.289 ^##^	1.381 ± 1.172 **	5.872	0.004
C21:0	0.111 ± 0.027	0.113 ± 0.026	0.102 ± 0.024	1.031	0.362
C22:0	0.234 ± 0.099	0.259 ± 0.056	0.209 ± 0.086	2.056	0.136
C23:0	0.086 ± 0.033	0.091 ± 0.022	0.100 ± 0.032	1.194	0.309
C24:0	0.843 ± 0.169	0.937 ± 0.319	1.015 ± 0.317	2.261	0.112
C14:1N5	1.179 ± 0.428	1.573 ± 0.492 *	1.677 ± 0.812 *	4.551	0.014
C16:1N7	11.553 ± 5.369	11.227 ± 2.719	9.270 ± 3.681	2.045	0.137
C17:1N7	0.504 ± 0.200	0.576 ± 0.239	0.670 ± 0.308	2.521	0.088
C18:1TN9	1.153 ± 0.280	1.455 ± 0.548	1.149 ± 0.834	2.031	0.139
C18:1N9	182.544 ± 38.631	175.740 ± 39.452	165.288 ± 44.211	1.039	0.36
C20:1N9	2.695 ± 0.928	2.592 ± 0.822	2.174 ± 0.685	2.554	0.085
C22:1N9	6.020 ± 1.330	2.985 ± 2.111 **	6.104 ± 2.437 ^&&^	18.178	<0.001
C24:1N9	14.558 ± 2.618	15.150 ± 3.552 ^##^	10.549 ± 2.793 **	15.481	<0.001
C18:3N3	11.857 ± 4.518	10.793 ± 4.559	9.269 ± 3.454	1.948	0.151
C20:3N3	0.587 ± 0.153	0.571 ± 0.285	0.511 ± 0.132	0.894	0.414
C20:5N3 (EPA)	11.573 ± 3.491	9.923 ± 2.362	11.056 ± 2.591 *	2.060	0.136
C22:6N3 (DHA)	15.966 ± 4.189	11.276 ± 3.182 **	12.777 ± 2.447 **	11.865	<0.001
C22:5N3 (DPA)	14.801 ± 3.149	12.519 ± 3.708 *	12.464 ± 2.448 *	4.083	0.021
C18:2N6	928.338 ± 206.304	1085.928 ± 418.170	1170.837 ± 291.339 *	3.354	0.041
C18:3N6	1.290 ± 0.456	2.789 ± 1.344 **	2.561 ± 1.049 **	14.976	<0.001
C20:2N6	3.687 ± 0.671	3.594 ± 0.631	3.219 ± 0.719	3.059	0.054
C20:3N6	14.085 ± 3.615	13.711 ± 4.045	12.383 ± 4.284	1.142	0.325
C20:4N6	69.490 ± 14.571	65.819 ± 13.499	58.883 ± 14.304 *	3.307	0.043
C22:2N6	0.605 ± 0.143	0.788 ± 0.226 **	0.557 ± 0.199 ^&&^	9.249	<0.001
C22:4N6	2.300 ± 0.805	2.840 ± 0.682 *	2.638 ± 0.559	3.651	0.031
C22:5N6	2.109 ± 0.664	2.397 ± 0.716	2.029 ± 0.615	1.916	0.155

Note: * *p* < 0.05 vs. HC group, ** *p* < 0.01 vs. HC group; ^#^ < 0.05 vs. SCH group, ^##^ < 0.01 vs. SCH group; ^&&^ < 0.01 vs. MDD group. Abbreviations: C8:0, caprylic acid; C10:0, capric acid; C12:0, lauric acid; C13:0, tritridecanoin; C14:0, myristic acid; C15:0, pentadecanoic acid; C16:0, palmitic acid; C17:0, heptadecanoic acid; C18:0, stearic acid; C20:0, arachidic acid; C21:0, heneicosanoic acid; C22:0, behenic acid; C23:0, tricosanoic acid; C24:0, lignoceric acid; C14:1N5, myristoleic acid; C16:1N7, palmitoleic acid; C17:1N7, margaric acid; 18:1n9, oleic acid; C18:1TN9, elaidic acid; C20:1N9, eicosaenoic acid; C22:1N9, erucic acid; C24:1N9, nervonic acid; C18:3N3, α-linolenic acid; C20:3N3 and C20:2N6, eicosatrienoic acid; C20:5N3, eicosapentaenoic acid (EPA); C22:5N3, docosapentaenoic acid (DPA); C22:6N3, docosahexaenoic acid (DHA); C18:2N6, linoleic acid; C18:3N6, gamma-linolenic acid; C20:3N6, gamma dihomo linoleic acid; C20:4N6, arachidonic acid; C22:2N6, docosadienoic acid; C22:4N6, docosatetratenoic acid; C22:5N6, docosapentanoic acid.

## Data Availability

Not applicable.

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
