# Peer review of "A Preliminary Comparison of Plasma Tryptophan Metabolites and Medium- and Long-Chain Fatty Acids in Adult Patients with Major Depressive Disorder and Schizophrenia"

_medicina, 2023, doi:10.3390/medicina59020413_

Round 1

Reviewer 1 Report (Previous Reviewer 3)

The revised version is considerably better. Some language problems should still be sorted out.  There are errors and  minor inaccuracies (e.g, figure captions should be improved).

Author Response

Sorry for the trouble caused by the language problem. Your suggestions are of great significance to improve the readability and scientificity of the article.

We revised the captions, carefully checked the full text and made corresponding revisions.

I hope these modifications can meet your requirements.

Reviewer 2 Report (Previous Reviewer 2)

I am satisfied with the work done, I have no comments

Author Response

Thank you for your review. It's my pleasure.

This manuscript is a resubmission of an earlier submission. The following is a list of the peer review reports and author responses from that submission.

Round 1

Reviewer 1 Report

Mental illnesses belong to the second most widespread diseases with a great impact on society and are therefore of interest to many scientific teams. The association of tryptophan metabolites via the kynurenine pathway with fatty acids is similarly of great scientific interest.

However, the overview presented work in its current form is not suitable for acceptance.

1. It is very difficult for the reader to orient yourself in the work due to the large number of abbreviations used, which are not always explained

2. A fundamental shortcoming is that the images lack an explanation of the abbreviations used in the text accompanying the image

3. When informing which examination scales for MDD and Sch were used, it would be appropriate for a non-psychiatric expert to state, e.g., that a higher HAMA or HAMD is a worse clinical condition (line 68)

4. Figure 1 is difficult to read. Perhaps it would be enough to list only the significant differences and put the non-significant ones in the table, and the part of figure 1 L is not readable at all

5. The importance of EPA (C20:5, n-3 FA) and DHA (C22:6, n-3 FA) is the most discussed in MDD. These FAs are not analyzed in the work?  Why?

6. Figure 2 (A - D) are unreadable and figure 2 E - is absolutely unreadable, even with the use of a strong magnifying glass

7. In the discussion in the paragraph about Trp metabolites (lines 241-271), the information is presented mixed - once for MDD, then for Sch, then again for MDD - I suggest editing it so that the reader can understand it

8. Similarly, I suggest modifying the paragraph with the results with fatty acids (lines 305-339) so that it is understandable for the reader, e.g. according to diagnosis, but not according to mix diagnoses

9. I strongly recommend inserting into the text illustrative schematics of methanolic pathways and sites of potential action of FA.

Author Response

Reply to the comments of Reviewer 1

Mental illnesses belong to the second most widespread diseases with a great impact on society and are therefore of interest to many scientific teams. The association of tryptophan metabolites via the kynurenine pathway with fatty acids is similarly of great scientific interest. 

However, the overview presented work in its current form is not suitable for acceptance. 

  1. It is very difficult for the reader to orient yourself in the work due to the large number of abbreviations used, which are not always explained

Response: Sorry for the nonstandard writing. Abbreviations are standardized in the revised manuscript.

  1. A fundamental shortcoming is that the images lack an explanation of the abbreviations used in the text accompanying the image

Response: Abbreviations are added in the legends of the image of revised manuscript.

  1. When informing which examination scales for MDD and Sch were used, it would be appropriate for a non-psychiatric expert to state, e.g., that a higher HAMA or HAMD is a worse clinical condition (line 68)

Response: We added the description “The scores were positively correlated with anxiety, depression and psychosis symptoms, respectively.” In line 71.

  1. Figure 1 is difficult to read. Perhaps it would be enough to list only the significant differences and put the non-significant ones in the table, and the part of figure 1 L is not readable at all

Response: Thank you for your suggestion, we revised Figure 1 and put the non-significant ones in table 2.

  1. The importance of EPA (C20:5, n-3 FA) and DHA (C22:6, n-3 FA) is the most discussed in MDD. These FAs are not analyzed in the work? Why?

Response: This is indeed one of the shortcomings of this study. This is a regrettable result, EPA and DHA were not detected out in blood samples in the present study. This may be related to the analysis method of the mass spectrometry, the stability of the standard and the plasma storage conditions. Meanwhile, in order to test all samples in the same batch, we put the collected blood samples into liquid nitrogen for freezing which may also affect these two substances.

  1. Figure 2 (A - D) are unreadable and figure 2 E - is absolutely unreadable, even with the use of a strong magnifying glass

Response: We revised Figure 2, reduced the composition of fatty acid molecules and correlation heat map, and put the detailed data of FAs in table 3.

  1. In the discussion in the paragraph about Trp metabolites (lines 241-271), the information is presented mixed - once for MDD, then for Sch, then again for MDD - I suggest editing it so that the reader can understand it

Response: We adjusted the order and made corresponding modifications, bringing forward the discussion part of MDD and then following with the discussion part of SCH.

  1. Similarly, I suggest modifying the paragraph with the results with fatty acids (lines 305-339) so that it is understandable for the reader, e.g. according to diagnosis, but not according to mix diagnoses

Response: We also adjusted the order and made corresponding modifications with the results for fatty acids, bringing forward the discussion part of MDD and then following with the discussion part of SCH, and putting the discussion of their common differences at the end.

  1. I strongly recommend inserting into the text illustrative schematics of methanolic pathways and sites of potential action of FA.

Response: That is a very important and meaningful work. But it is really too difficult for us due to our insufficient knowledge reserves. The metabolic pathway of fatty acids in central diseases has been reported, but the metabolic pathway of fatty acids in MDD and SCH still needs further investigation. We added this shortcoming in the last paragraph of the Discussion.

Reference:

Bogie, J.F.J., et al., Fatty acid metabolism in the progression and resolution of CNS disorders. Adv Drug Deliv Rev, 2020. 159: p. 198-213.

Reviewer 2 Report

The problem of psychiatric diseases is understudied and too little progress has been made in this field. I consider research and the search for new diagnostic markers of psychiatric diseases to be urgent.

This article is made at a sufficiently good level in terms of the quality of the text (excluding the "Conclusions"), the applied methods (although limited to the study of plasma only), and the illustration of the results.

On the other hand, there are significant remarks regarding:

1) novelties (61 studies comprising 2813 patients and 2948 healthy controls were devoted to this problem according to Abbas F Almulla, 2022)

2) the number of involved patients (insufficient, 4.5 times lower than the average research group in this field)

3) the "Conclusions" section is poorly written, in fact it has no content

Author Response

Reply to the comments of Reviewer 2

The problem of psychiatric diseases is understudied and too little progress has been made in this field. I consider research and the search for new diagnostic markers of psychiatric diseases to be urgent. This article is made at a sufficiently good level in terms of the quality of the text (excluding the "Conclusions"), the applied methods (although limited to the study of plasma only), and the illustration of the results. On the other hand, there are significant remarks regarding:

1) novelties (61 studies comprising 2813 patients and 2948 healthy controls were devoted to this problem according to Abbas F Almulla, 2022)

Response: The composition of tryptophan catabolite in schizophrenia and MDD has been well-determined in previous work. The innovation of this study is only to compare the differences between patients with depression and schizophrenia. Meanwhile, the present study also compared the differences in fatty acid composition between the two groups.

2) the number of involved patients (insufficient, 4.5 times lower than the average research group in this field)

Response: This is indeed one of the shortcomings of this study. This is indeed one of the shortcomings of this study. This makes the results of this study need a large sample of clinical research to verify in the future.

3) the "Conclusions" section is poorly written, in fact it has no content

Response: That is a very good suggestion. We re-write the Conclusion as “In summary, the present study characterized plasma Trp metabolites and the composition of MCFAs and LCFAs in adult patients with MDD and SCH. Trp and NKY as well as total PUFA and N6 PUFA were decreased in both patients with MDD and SCH. Furthermore, combinatorial classifiers that could be able to discriminate MDD from HC or SCH from HC, as well as MDD from SCH were developed. Intriguingly, UFA may play a role to distinguish the two diseases. Our findings provide a potential strategy to combine Trp with FA metabolism for the diagnosis of MDD and SCH.”

Reviewer 3 Report

This sufficiently good paper should also reference relevant recent publications that deal with the role of the microbiota in producing neurotransmitters in the human organism, both in health and disease, e.g., Oleskin, A. V. and Shenderov, B. A. (2020). MICROBIAL COMMUNICATION AND MICROBIOTA-HOST INTERACTIONS: BIOMEDICAL, BIOTECHNOLOGICAL, AND BIOPOLITICAL IMPLICATIONS.. Some language-editing is recommendable  

Author Response

Reply to the comments of Reviewer 3

This sufficiently good paper should also reference relevant recent publications that deal with the role of the microbiota in producing neurotransmitters in the human organism, both in health and disease, e.g., Oleskin, A. V. and Shenderov, B. A. (2020). MICROBIAL COMMUNICATION AND MICROBIOTA-HOST INTERACTIONS: BIOMEDICAL, BIOTECHNOLOGICAL, AND BIOPOLITICAL IMPLICATIONS..

Some language-editing is recommendable 

Response: Thank you for your suggestion. We modified the language in the revised manuscript. We also added the description “Nevertheless, both fatty acids and Trp metabolism were regulated by gut microbiota, the metabolic pathway of fatty acids in central nervous system diseases and the composition characteristics of gut microbiota in depression and schizophrenia have been clarified [50, 61, 62]. The relationship between specific gut microbiota and fatty acid metabolism and tryptophan metabolism also needs to be further explored.” In the paragraph of Discussion.

Round 2

Reviewer 1 Report

The main problem is that the authors did not provide answers to the reviewer, or a comment on the reviewer's opinion. The authors included explanations of the used abbreviations in the text to the figures, but did not explain all of them, or gave explanations for those abbreviations that are not used in the figure. This indicates inconsistent work.

The authors did not explain their findings, which are in conflict with current findings. We compared the concentrations of some FA listed in the table, e.g. C18:2N6 in a healthy control - 325.595 ng/mL ± 71.535, but our controls had values in the 500-600 ug/mL range, and so did others.

It is not explained why the authors found that in MDD, the dominant pathophysiological feature is a decrease in n-6 FA and not n-3, which correlate with depressive symptom severity, as reported in a large number of publications and reviews.

I stand by my original opinion.

Author Response

Reply to the comments of Reviewer 1

The main problem is that the authors did not provide answers to the reviewer, or a comment on the reviewer's opinion. The authors included explanations of the used abbreviations in the text to the figures, but did not explain all of them, or gave explanations for those abbreviations that are not used in the figure. This indicates inconsistent work.

Response: We are shameful for the misunderstanding last time, especially the abbreviations of fatty acids were missed. We added the description “Abbreviations: MDD, Major depressive disorder; SCH, schizophrenia; HC, healthy controls; Trp, tryptophan; FA, fatty acids; medium-chain fatty acids, MCFAs; long -chain fatty acids, LCFAs; NKY, N-formyl-kynurenine; 5-HIAA, 5-Hydroxyindole-3-acetic acid; UFA, unsaturated fatty acids; PUFA, polyunsaturated fatty acids; N3, Omega-3 (N3) polyunsaturated fatty acids; N6, Omega-6 (N6) polyunsaturated fatty acids; LC-MS, liquid chromatography-mass spectrometry; GC-MS, gas chromatography-mass spectrometry; ROC, receiver operating characteristic; DSM-5, Diagnostic and statistical manual of mental disorders, fifth edition; HAMA, Hamilton Anxiety Scale; HAMD, Hamilton Depression Rating Scale ; PANSS, Positive and Negative Syndrome Scale; BMI, body mass index; SD, standard deviation; PANSS (T), PANSS total score; PANSS (P), PANSS positive symptom score; PANSS (N), PANSS negative symptom score; PANSS (G), PANSS general psychopathological symptom score; ANOVA, analysis of variance; IAId, Indole-3-carboxaldehyde; IAA, Indoleacetate; ILA, Indole-3-lactic acid; Kyn, L-kynurenine; PIC, Picolinic acid; QUIN, Quinolinic acid; I3S, 3-Indoxyl sulfate; SFA, saturated fatty acids; UFA, unsaturated fatty acids; C8:0, Caprylic acid; C10:0, Capric acid; C12:0, Lauric acid; C13:0, Tritridecanoin; C14:0, Myristic acid; C15:0, Pentadecanoic acid; C16:0, Palmitic acid; C17:0, Heptadecanoic acid; C18:0, Stearic acid; C20:0, Arachidic acid; C21:0, Heneicosanoic acid; C22:0, Behenic acid; C23:0, Tricosanoic acid; C24:0, Lignoceric acid; C14:1N5, Myristoleic acidï¼›C16:1N7, Palmitoleic acid; C17:1N7, Margaric acid; 18:1n9, Oleic acid; C18:1TN9, Elaidic acid; C20:1N9, Eicosaenoic acid; C22:1N9, Erucic acid; C24:1N9, Nervonic acid; C18:3N3, α-linolenic acid; C20:3N3 and C20:2N6, Eicosatrienoic acid; C22:5N3, Docosapentaenoic acid; C18:2N6, Linoleic acid; C18:3N6, Gamma-linolenic acid; C20:3N6, Gamma dihomo linoleic acid; C20:4N6, Arachidonic acid; C22:2N6, Docosadienoic acid; C22:4N6, Docosatetratenoic acid; C22:5N6, docosapentanoic acid.” after Keywords. In addition, we also added the abbreviations in the table notes of Table 3 and legend of Figure 3.

The authors did not explain their findings, which are in conflict with current findings. We compared the concentrations of some FA listed in the table, e.g. C18:2N6 in a healthy control - 325.595 ng/mL ± 71.535, but our controls had values in the 500-600 ug/mL range, and so did others.

Response: Thank you for your careful review. What makes us ashamed is that the unit in Table 3 is incorrect. Actually, the unit of parameter detected in table 2 was ng/mL whereas parameter detected in table 3 was ng/mL. We regret again for carelessness in revising the manuscript. Meanwhile, it is also a very important question that the fatty acid content of blood samples detected in different studies was not consistent. This inconsistency may be related to the factors, such as eating habits, gender and age. Of note, the influence of plasma collection methods and storage conditions cannot be ignored. We added the description “Furthermore, several fatty acid contents detected in the present study was not con-sistent with previous work. For example, the concentrations of C14:0, C18:0 and C22:0 was inconsistent with other studies [72, 73]. This inconsistency may be related to the eating habits, living standards of the participants, gender and age. Of note, the influence of plasma collection methods, storage conditions and fatty acid detection and analysis methods on the results cannot be ignored [74].” in the last paragraph of Discussion.

References:

  1. Qureshi, W., et al., Risk of diabetes associated with fatty acids in the de novo lipogenesis pathway is independent of insulin sensitivity and response: the Insulin Resistance Atherosclerosis Study (IRAS). BMJ Open Diabetes Res Care, 2019. 7(1): p. e000691.
  2. Chiu, H.H., et al., An efficient and robust fatty acid profiling method for plasma metabolomic studies by gas chromatography-mass spectrometry. Clin Chim Acta, 2015. 451(Pt B): p. 183-90.
  3. Jackson, K.H., et al., Plasma fatty acid responses to a calorie-restricted, DASH-style diet with lean beef. Prostaglandins Leukot Essent Fatty Acids, 2022. 179: p. 102413.

It is not explained why the authors found that in MDD, the dominant pathophysiological feature is a decrease in n-6 FA and not n-3, which correlate with depressive symptom severity, as reported in a large number of publications and reviews.

Response: That is a very important question. We discussed this inconsistent result and added the description as “It should be noted that a number of studies found that N3 but not N6 was decreased in MDD. For example, levels of N3 were significantly lower in depressive patients and there was no significant change in N6 between MDD and control subjects [53], and lower levels of total N3 and increased N6/N3 ratios were also reported in perinatal depression patients [57]. However, a recent study found that N3 was lower in depression, but it was not consistently associated with subsequent change in depressive symptoms [58]. Meanwhile, circulating PUFAs were unlikely to reflect a vulnerability marker for the recurrence of depression [59]. Another study further indicated that the N6:N3 ratio was positively associated with MDD at age 24 but there was little evidence of cross-sectional associations between PUFA measures and mental disorders at age 17 [60]. On the other hand, dietary supplementation with N3 during pregnancy or postpartum reduced depressive symptoms and N3 adjuvant treatment was a potential option for depression and anxiety symptoms of people with recent onset psychosis [61, 62]. However, recent studies found that N3 have an overall significant small beneficial effect on perinatal depression [63], and the evidence of N3 supplementation on sertraline continuous therapy to reduce depression or anxiety symptoms is not enough to make recommendations [64]. Moreover, other studies also found that N3 supplementation probably has little or no effect in preventing depression or anxiety symptoms [65] and non-clinically beneficial effect of N3 on depressive symptomology when compared to placebo in adult patients with depression [66]. Furthermore, basic study already showed that complementation of the diet with arachidonic acid (AA, C20:4N6) was sufficient to alleviate both the microbiota-induced depressive-like behaviors [67]. Together, the changes of PUFA in patients with depression in different studies are not completely consistent. Of note, several N3 was not detected out in the present study, which will lead to a decrease in the overall content of N3 and an increase in the N6: N3 ratio. Therefore, the results of this study need to be further verified after the improvement of the accuracy for FA detection technology.” at the end of the 6th paragraph in Discussion.

References:

  1. Lin, P.Y., S.Y. Huang, and K.P. Su, A meta-analytic review of polyunsaturated fatty acid compositions in patients with depression. Biol Psychiatry, 2010. 68(2): p. 140-7.
  2. Lin, P.Y., et al., Polyunsaturated Fatty Acids in Perinatal Depression: A Systematic Review and Meta-analysis. Biol Psychiatry, 2017. 82(8): p. 560-569.
  3. Thesing, C.S., et al., Bidirectional longitudinal associations of omega-3 polyunsaturated fatty acid plasma levels with depressive disorders. J Psychiatr Res, 2020. 124: p. 1-8.
  4. Thesing, C.S., et al., Fatty acids and recurrence of major depressive disorder: combined analysis of two Dutch clinical cohorts. Acta Psychiatr Scand, 2020. 141(4): p. 362-373.
  5. Mongan, D., et al., Plasma polyunsaturated fatty acids and mental disorders in adolescence and early adulthood: cross-sectional and longitudinal associations in a general population cohort. Transl Psychiatry, 2021. 11(1): p. 321.
  6. Robinson, D.G., et al., A potential role for adjunctive omega-3 polyunsaturated fatty acids for depression and anxiety symptoms in recent onset psychosis: Results from a 16 week randomized placebo-controlled trial for participants concurrently treated with risperidone. Schizophr Res, 2019. 204: p. 295-303.
  7. Hsu, M.C., C.Y. Tung, and H.E. Chen, Omega-3 polyunsaturated fatty acid supplementation in prevention and treatment of maternal depression: Putative mechanism and recommendation. J Affect Disord, 2018. 238: p. 47-61.
  8. Mocking, R.J.T., et al., Omega-3 Fatty Acid Supplementation for Perinatal Depression: A Meta-Analysis. J Clin Psychiatry, 2020. 81(5).
  9. Chambergo-Michilot, D., et al., Efficacy of omega-3 supplementation on sertraline continuous therapy to reduce depression or anxiety symptoms: A systematic review and meta-analysis. Psychiatry Res, 2021. 296: p. 113652.
  10. Deane, K.H.O., et al., Omega-3 and polyunsaturated fat for prevention of depression and anxiety symptoms: systematic review and meta-analysis of randomised trials. Br J Psychiatry, 2021. 218(3): p. 135-142.
  11. Appleton, K.M., et al., Omega-3 fatty acids for depression in adults. Cochrane Database Syst Rev, 2021. 11(11): p. CD004692.
  12. Chevalier, G., et al., Effect of gut microbiota on depressive-like behaviors in mice is mediated by the endocannabinoid system. Nat Commun, 2020. 11(1): p. 6363.
  13. Deane, K.H.O., et al., Omega-3 and polyunsaturated fat for prevention of depression and anxiety symptoms: systematic review and meta-analysis of randomised trials. Br J Psychiatry, 2021. 218(3): p. 135-142.
  14. Appleton, K.M., et al., Omega-3 fatty acids for depression in adults. Cochrane Database Syst Rev, 2021. 11(11): p. CD004692.
  15. Chevalier, G., et al., Effect of gut microbiota on depressive-like behaviors in mice is mediated by the endocannabinoid system. Nat Commun, 2020. 11(1): p. 6363.

Reviewer 2 Report

The article is sufficiently well prepared and has some scientific novelty. Scientific value can occur if the research is presented as a pilot project or a preliminary study. (number of involved patients).

Author Response

Reply to the comments of Reviewer 2

The article is sufficiently well prepared and has some scientific novelty. Scientific value can occur if the research is presented as a pilot project or a preliminary study. (number of involved patients).

Response: Thank you for your suggestion. The conclusion of our manuscript is inappropriate dur to the limited enrolled patients. Thus, the title was instead by “A preliminary comparison of plasma tryptophan metabolites and medium and long chain fatty acids in adult patients with major depressive disorder and schizophrenia”. Meanwhile, “may” in the conclusions of Abstract was instead by “might”.

Round 3

Reviewer 1 Report

Comment to some answers of authors:

First answers:

  1. The importance of EPA (C20:5, n-3 FA) and DHA (C22:6, n-3 FA) is the most discussed in MDD. These FAs are not analyzed in the work? Why?

Response: This is indeed one of the shortcomings of this study. This is a regrettable result, EPA and DHA were not detected out in blood samples in the present study. This may be related to the analysis method of the mass spectrometry, the stability of the standard and the plasma storage conditions. Meanwhile, in order to test all samples in the same batch, we put the collected blood samples into liquid nitrogen for freezing which may also affect these two substances.

In the case of the analysis of fatty acids in patients with a mental disorder, the determination of omega-3 FA is necessary, as their optimal level in the brain/serum is currently considered responsible for suppressing the basic pathophysiological features of depressive disorders and schizophrenia. Such features include an inflammatory reaction, a disorder in the metabolism of neurotransmitters, as well as membrane fluidity. Conversely, omega-6 FAs trigger a significant pro-inflammatory response. For this reason, the ratio of omega-6/omega-3 FA is significant and is even more significant than the concentrations of EPA and DHA alone. The positive correlation between the severity of depression and the omega-6/omega-3 ratio clearly demonstrates this (Trebatická J. et al., Psychiatry Research, 2020). These relationships are currently sufficiently proven.

Therefore, the work, which, even if it failed to determine omega-3 FA from a methodological point of view, cannot meet the demanding criteria for publication in Medicina.

Second answers:

The authors did not explain their findings, which are in conflict with current findings. We compared the concentrations of some FA listed in the table, e.g. C18:2N6 in a healthy control - 325.595 ng/mL ± 71.535, but our controls had values in the 500-600 ug/mL range, and so did others.

Response: Thank you for your careful review. What makes us ashamed is that the unit in Table 3 is incorrect. Actually, the unit of parameter detected in table 2 was ng/mL whereas parameter detected in table 3 was ng/mL. We regret again for carelessness in revising the manuscript. Meanwhile, it is also a very important question that the fatty acid content of blood samples detected in different studies was not consistent. This inconsistency may be related to the factors, such as eating habits, gender and age. Of note, the influence of plasma collection methods and storage conditions cannot be ignored. We added the description “Furthermore, several fatty acid contents detected in the present study was not con-sistent with previous work. For example, the concentrations of C14:0, C18:0 and C22:0 was inconsistent with other studies [72, 73]. This inconsistency may be related to the eating habits, living standards of the participants, gender and age. Of note, the influence of plasma collection methods, storage conditions and fatty acid detection and analysis methods on the results cannot be ignored [74].” in the last paragraph of Discussion.

I agree with the part of the answer where the authors state that the results of fatty acid concentrations from individual laboratories may differ depending on the methods used, or also depending on sample pre-treatments, sample storage methods, on the diet of the patients, etc., etc. ... 

However, I have learned reservations about the comments on the second answe:

I agree with the authors' comment regarding cited literature 53-57.

However, the cited works (58-61) discuss the effect of omega-3 FA after their supplementation, while the authors' work evaluates the composition of FA without supplementation. I agree with the authors that the results of omega-3 FA supplementation are not clear cut. However, meta-analyses as well as reviews point to a possible cause of failure of adjuvant treatment with omega-3 FA (63-66), namely a different design, different dose of the supplement, time of administration, wrong diagnosis by the patient (the correctness of the diagnosis, like major depression and depression with some comorbidities), etc...

In this case, the authors do not answer the question. In response, the authors mix data on serum fatty acids in patients without fatty acid supplementation and with supplementation, which are two quite different situations.

At the end of the second answer, the authors confirm that they did not determine enough omega-3 FA, therefore "the results of this study need to be further verified after the improvement of the accuracy for FA detection technology."

For this reason, I recommend that they first use verified methods for determining serum fatty acids, evaluate them correctly, and then publish results in a journal with a good impact factor